# Calcific Shoulder Tendinopathy (CT): Influence of the Biochemical Process of Hydrolysis of HA (Hydroxyapatite) on the Choice of Ultrasound-Guided Percutaneous Treatment (with the Three-Needle Technique)

Stefano Galletti [1], Marco Miceli [1], Salvatore Massimo Stella [2], Fabio Vita [1], Davide Bigliardi [3], Danilo Donati [1], Domenico Creta [4] and Antonio Frizziero [3,*]

1    IRCCS-Rizzoli Orthopedic Institute, University of Bologna, 40136 Bologna, Italy; stefanogalletti2011@gmail.com (S.G.); marco.miceli@ior.it (M.M.); vitafabio@hotmail.it (F.V.); danilo.donati@ior.it (D.D.)
2    School of Musculoskeletal Ultrasound, University of Pisa, 56126 Pisa, Italy; smstella@alice.it
3    Physical and Rehabilitation Medicine, University of Parma, 43126 Parma, Italy; davide.bigliardi@unipr.it
4    Physical Therapy and Rehabilitation Service, Private Hospital "Madre FortunataToniolo", 40136 Bologna, Italy; domenico.creta@alice.it
*    Correspondence: antonio.frizziero@unipr.it

**Abstract:** Calcific shoulder tendinopathy (CT) is a common condition involving the central part or insertion of the rotator cuff tendons (RC) or the subacromial-subdeltoid bursa (SASD). The calcific deposits consist of poorly crystallized calcium hydroxyapatite but the mechanism of their formation still remains unclear. CT can be divided into three distinct stages, as reported by Uthhoff et al. Clinically, this condition varies with the extent of the calcification and the phase of the condition. In particular, the disorder is asymptomatic or may cause mild discomfort during the deposition of calcium, while it becomes acutely painful during the resorptive phase. US-PICT (ultrasound-guided percutaneous irrigation of calcific tendinopathy) is indicated in the acute phase (resorptive phase) of CT with significant pain relief and a very low rate of minor complications. The aim of this manuscript is to define the rationale of the ultrasound-guided percutaneous irrigation of calcific tendinopathy, correlating it with the sequence of biochemical processes that lead to the hydrolysis of hydroxyapatite. Furthermore, we will explain the reasons why we prefer using the three-needle technique for the dissolution of calcifications.

**Keywords:** calcific shoulder tendinopathy; hydroxyapatite; infiltrative technique; ultrasound; US-PICT





## 1. Introduction

Calcific shoulder tendinopathy (CT) is a common condition involving the central part or insertion of the rotator cuff tendons (RC) or the subacromial-subdeltoid bursa (SASD). CT occurs in 2.5–7.5% of healthy shoulders in adults, more frequently in women than in men, especially in the 4th or 5th decade of life [1,2].

The most affected area of the RCis the critical zone of the supraspinatus tendon (80%), followed by the infraspinatus (15%) and the subscapularis tendon (5%) [3,4]; in particular, the critical area of the supraspinatus tendon is a hypovascular region located about 1 cm from the tendoninsertion on the greater humeral tuberosity, which is consequently more vulnerable to diseases such as tendinopathies and calcifications [5,6].

The calcific deposits consist of poorly crystallized calcium hydroxyapatite [2,7,8] but the mechanism of their formation still remains unclear.

CT can be divided into three distinct stages, as reported by Uthhoffet al. in 1997:

-    In the precalcific stage, a portion of the tendon undergoes a fibrocartilaginous transformation, which acts as a substrate for calcium deposition and is usually asymptomatic.

- The calcific phase, which is further divided into three substages: the formative phase, resting phase and resorptive phase. In the formative phase occurs a calcium crystal deposition in the transformed tissue, which is mediated by the chondrocytes of the fibrocartilaginous metaplasia, which is usually asymptomatic for a long time; in the resting phase, the calcific deposits remain stable; in the resorptive phase, vascular tissue develops around the calcific deposit and macrophages and multinucleated giant cells hydrolyse and phagocyte the calcific deposit. Calcifications occasionally leaks into adjacent structures (more frequently SASD bursa). This phase is usually associated with the development of acute pain.
- In the post-calcific stage, the calcium deposits are replaced by granulation tissue [6,9].

Clinically, this condition varies with the extent of the calcification and the phase of the condition, as described above.

In particular, the disorder is asymptomatic or may cause mild discomfort during the deposition of calcium, while it becomes acutely painful during the resorptive phase [5,8,10–12].

Diagnostic imaging is necessary to distinguish CT from other painful subacromial conditions.

Conventionalradiography may detect calcification in the soft tissue around the humerus and in the subacromial space.

An ultrasound (US) is agold standard exam as it can detect calcifications with a high sensitivity and allows one to highlight four different calcificationpatterns:

- Type I: hyperechoic calcification with well-defined posterior acoustic shadowing due to their consistent amount of calcium (Figure 1A).
- Type II: hyperechoic calcification with mild posterior acoustic shadowing, due to the reduced amount of calcium as a consequence of hydrolysis from the hydrolysis of the calcium deposits (Figure 1B).
- Type III: isoechoic calcification without posterior acoustic shadowing, due to the softening of the deposit (Figure 1C).
- Type IV: fluid hypo/anechoic calcifications without posterior acoustic shadowing, which can bring the formation of a pseudo-abscess (Figure 1D,E).

The first and second types correspond to the precalcifying and formative phases while the last two types are typical of the resorptive phase. Due to their fluid nature, type 3 and 4 calcifications can migrate from one site to another, as mentioned above, usually in the direction of the subacromial-subdeltoid bursa (SASD) or towards the synovial fluid in the joint cavity or directly in the bone, through osteoclastic activation [1,13].

CT treatment is initially conservative and usually involves rest, physical therapy (in particular, high-energy extracorporeal shock waves [14]) and oral non-steroidal anti-inflammatory drugs (NSAIDs) administration. In the case of the failure of the first-line approach, an infiltrative treatment may be performed, using or not using US guidance [1,4].

US-PICT (ultrasound-guided percutaneous irrigation of calcific tendinopathy) is indicated in the acute phase of CT with significant pain relief and a very low rate of minor complications such as a vaso-vagal reaction and bursitis [4]. The treatment is performed with needling and by washing the calcifications with a 0.9% sodium chloride saline solution, and can be performed with various personal techniques using one, two or three needles to inject and retrieve the fluid to dissolve the calcium deposits, depending on the friability or the degree of the softening of the deposit. Recently, this technique has also been performed in calcific tendinopathies outside the rotator cuff with excellent results in terms of pain reduction and confirming its safety [15].

Surgery (arthroscopy) is considered the last treatment option in chronic cases where conservative or less invasive approaches have failed [1,4,16].

The aim of this manuscript is to define the rationale of the ultrasound-guided percutaneous irrigation of calcific tendinopathy, correlating it with the sequence of biochemical processes that lead to the hydrolysis of hydroxyapatite. Furthermore, we will explain the reasons why we prefer to use the three-needle technique for the dissolution of calcifications.

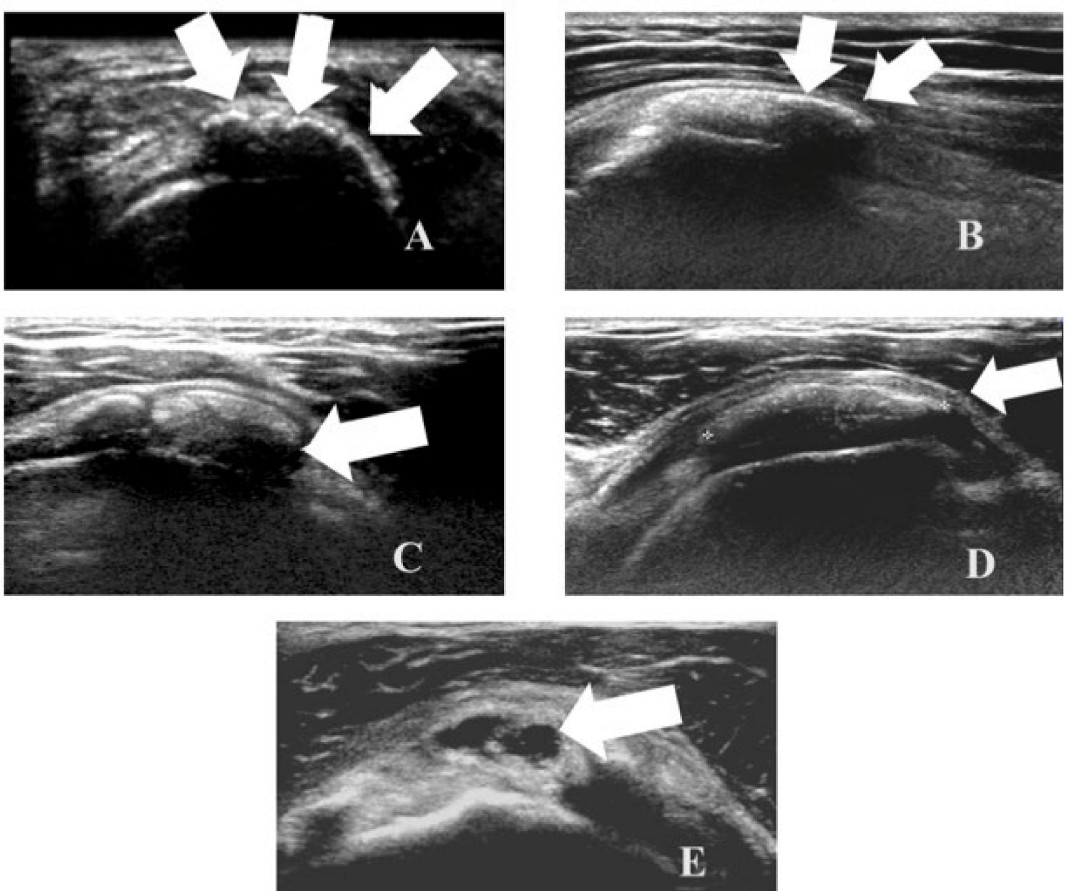

**Figure 1.** Four different ultrasound patterns of calcific tendinopathy: hyperechoic calcification with well-defined posterior acoustic shadows(**A**); hyperechoic calcification with faint posterior acoustic shadows (**B**); isoechoic calcification with absent posterior acoustic shadows (**C**); hypoechoic calcification with absent posterior acoustic shadows that can become liquid and bring to pseudo-abscess phase (**D,E**).

## 2. Exposition of the Theory of the Formation and Dissolution of Calcific Tendinopathy

The etiopathogenesis of calcific tendinopathy is still controversial, mainly because it remains difficult to elucidate the primum movens causing the development of the calcifications, although it seems to be the result of an active cell-mediated process.

Following acute injury, repeated microtrauma and chemical injuries that damage the tendon, the natural healing process is initiated, which involves many sequential processes such as the synthesis and remodelling of the matrix, synthesis of pro-inflammatory cytokines, neovascularization, neural modulations and the recruitment of multi-potent cells. The occurrence of alterations in each of the different phases of healing can lead to different combinations of histopathological changes.

The mis-differentiation of tendon progenitor cells into chondrocytes or osteoblasts-may contribute to the pathogenesis of CT, although the mechanism leading to this mis-differentiation is not fully understood. Probably factors such as the expression of BMPs, biglycan, fibromodulin and anunfavourable microenvironment induced by overuse play a key role [17]. Recently, Maffulli et al. identified several genes whose expression is altered in frankly pathological areas of the supraspinatus tendon of patients with calcific tendinopathy. Specifically, the expression of BMP4 and BMP6 mRNAs are significantly decreased in calcific areas, whereas there is an increase in cathepsin K mRNA, tTG2 mRNA and osteopontin in calcific tissue. The increased expression of osteopontin and tTG2 could be compatible with an increased production in the calcific area by osteoclast-like cells involved

in the resorption phase. On the other hand, a downregulation of BMPs 4 and 6 could be associated with a reduction in cellularity in this area or could be one of the causes [18].

Calcifications consist of hydroxyapatite crystals and a recent review made an attempt to create a general description of the dissolution process of apatite at the atomic (ionic) level [19].

The literature reports a wide range of risk factors involved in the onset of calcific rotator cuff tendinitis, such as female sex, diabetes mellitus, dyslipidemia, hypothyroidism and other endocrine disorders [20], and in our experience a crucial role in the development of this condition is played by repetitive overhead movements and microtrauma, although individuals involved in strenuous manual labour or athletic activities are no more commonly affected than those leading sedentary lives. (Figure 2A,B) [21].

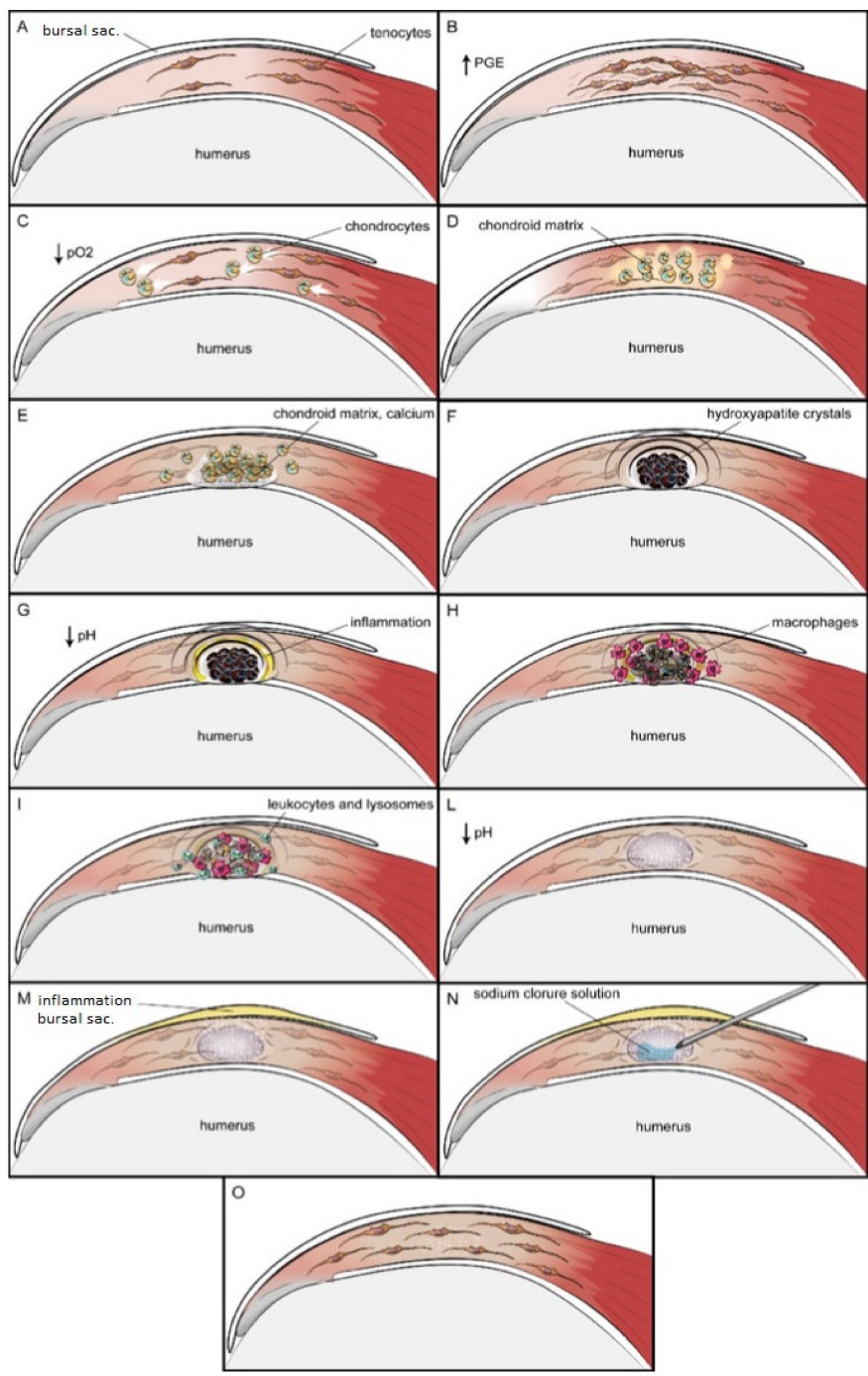

**Figure 2.** Exposition of the theory of the formation and dissolution of calcific tendinopathy: Repeated physical efforts or microtrauma are the initial factors of pathology (**A**,**B**).Mechanical stress with consequent

reduction in tissue Po2 can lead totenocytic metaplasia and subsequent deposition of the chondroid matrix ed calcium (**C–E**), which is associated with the reduction in the extracellular pH and leads to the formation of hydroxyapatite crystals, with manifestation on ultrasound imaging of ahypoechoic central zone and a posterior acoustic shadow (**E–G**). Subsequently, the cell-mediated attack of calcification byphagocytic macrophages, leukocytes and lysosomes occurs with an attempt to break down the calcification (**H,I**).Calcification rises according to a pressure gradient, causing inflammation of the subacromial-subdeltoid bursa (**L**). All thisincreases solubility of hydroxyapatite, represented in ultrasound with the loss of the posterior acoustic shadow and anincrease in central hyperechogeneity (**M**). Subsequently, the calcification is washed with 0.9% chloride saline with areduction in the size of the calcification (**N,O**).

Repetitive mechanical stresses lead to an increase in PGE 2 (prostagladin E2) production by tendon fibroblasts, which is a mediator of acute inflammation and tendinopathy [22,23].

It is possible that the inflammatory state causes a reduction in Po2 in the tendon tissue, a modification of the gene expression and a metaplastic transformation of tendon stem cells (TSCs) in non-tenocytic cells such as chondrocytes, osteocytes or adipocytes [22,24,25] withsubsequent calcification inside the tendon (formative phase), characterized on ultrasound examinations by well-defined posterior acoustic shadowing (Figure 2C–F).

After the resting phase, which can last for several years, the resorptive phase begins with the rate of the biochemical degradation of the hydroxyapatite crystals by macrophages and multinucleated giant cells depending on the local inflammatory state, resulting in reduced extracellular pH (acidosis) (Figure 2G–I) [26,27].

During this phase, calcific deposits have a toothpaste-like appearance with a central hyper exogeneityand an absence of posterior acoustic shadowing on US examinations (Figure 2M).

During the resorptive phase, patients may present clinically severe pain and a very limited range of motion caused by the leaking of calcifications into adjacent structures with SASD bursitis development (Figure 2L). During this phase, it is possible to conductneedling and a washing of the calcifications with a 0.9% sodium chloride saline solution, which has the dual effect of removing calcium deposits and restoring the pH to physiological values, resulting in a reduction in the chemical irritation of the bursa and painful symptoms (Figure 2N,O) and thus achieving maximum effectiveness of this treatment due to the biochemical characteristics of the calcifications.

## 3. Description of the Three-Needle Infiltrative Technique in Calcific Tendinopathy

Several studies have indicated different approaches to CT treatment, all of which include the use of saline to dissolve calcium deposits using either one needle [28] or two needles [29] depending on the choice of the sonographer. In addition, recent studies have shown that a dual-needle approach is more appropriate for the treatment of large deposits whereas a single needle may be more useful in the treatment of fluid calcifications [30].

In particular, US-PICT (percutaneous ultrasound-guided irrigation of calcific tendinopathy) is indicated in symptomatic patients with a calcification greater than 7 mm, an ultrasound appearance of type 3 or 4 [30] and no rotator cuff tears (Table 1); on the other hand, it is contraindicated in asymptomatic patients, in calcifications $\leq$7 mm, in hyperechogenic calcifications with a posterior acoustic shadow on type 1 or 2 at the US exam and in the presence of rotator cuff tears.

**Table 1.** Indications and contraindications for US-PICT treatment.

| Indications | Contraindications |
|---|---|
| Type 3 or 4 in the US exam | Type 1 or 2 in the US exam |
| Calcification >7 mm | Calcification <7 mm |
| No cuff tears | Cuff tears |
| Symptomatic patient | Asymptomatic patient |

US-PICT has become the gold standard for CT treatment as it does not require patient sedation, hospitalisation or post-procedure immobilisation and it is also cost-effective and does not require rest from work [28–32].

Prior to treatment, calcification is identified by the ultrasound with a 5–13 MHz linear probe and any contraindications such as a rotator cuff tear are excluded; in particular, the supraspinatus tendon is examined in internal rotation with the hand behind the back, the infraspinatus tendon is visualised by placing the hand on the contralateral shoulder and the subscapularis tendon is examined in external rotation [33].

For the intervention, the patient is placed into an oblique lateral decubitus position opposite of the affected side, preferred to the sitting position, with the arm rotated to help expose the calcification. After sterile preparation of the skin and equipment, local anaesthesia is practiced in ultrasound guidance at three levels: subcutaneous, bursal and sub-bursal, or even intracalcific (Figure 3).

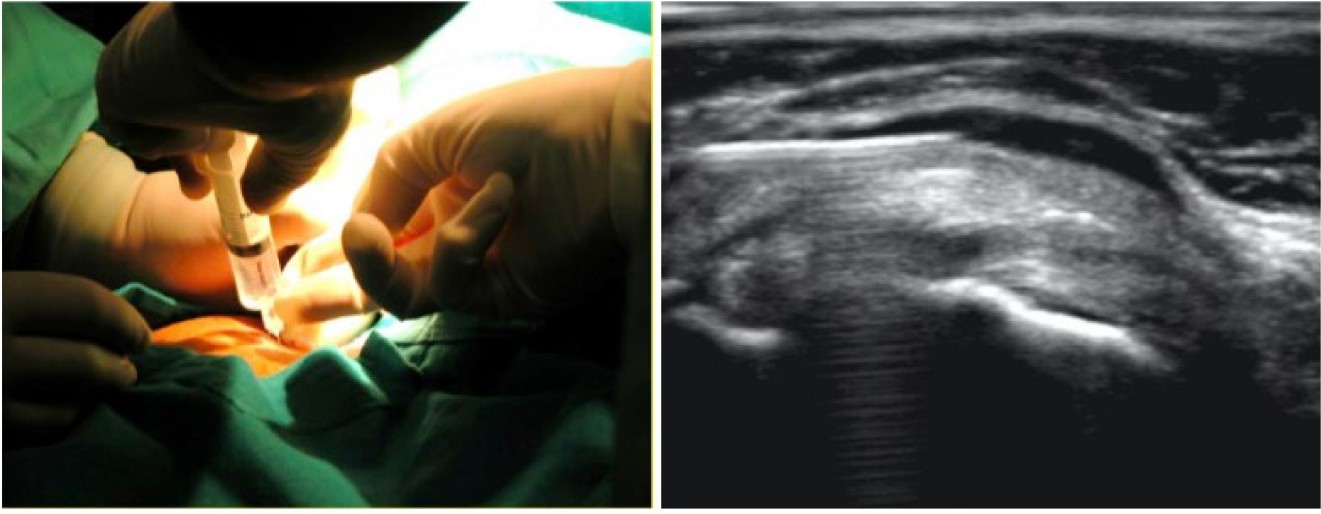

**Figure 3.** Performing ultrasound-guided bursal anaesthesia with lidocaine 1%.

We used a three-needle technique, where two large 18- or 16-gauge needles were inserted into the calcification (Figure 4).

The first is connected to a 60 mL syringe and a 500 mL bag of 0.9% sodium chloride saline.A second needle is inserted with the first into the calcification to aspirate its contents.

Subsequently, a third needle was inserted to facilitate the removal of the more resistant calcium formations (Figure 5).

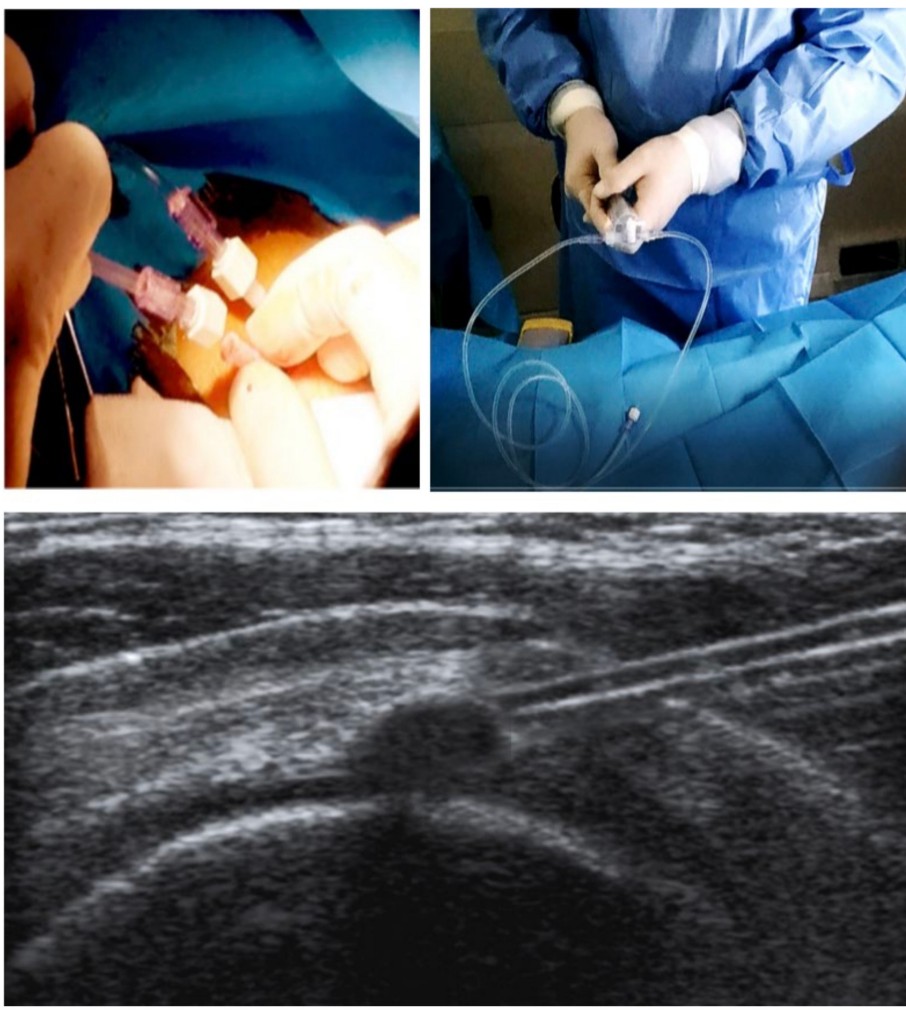

**Figure 4.** Two needles of 18 or 16 gauge are inserted into the calcification and are connected by two extensions joined by a three-way tap connected to a 60 mL syringe.

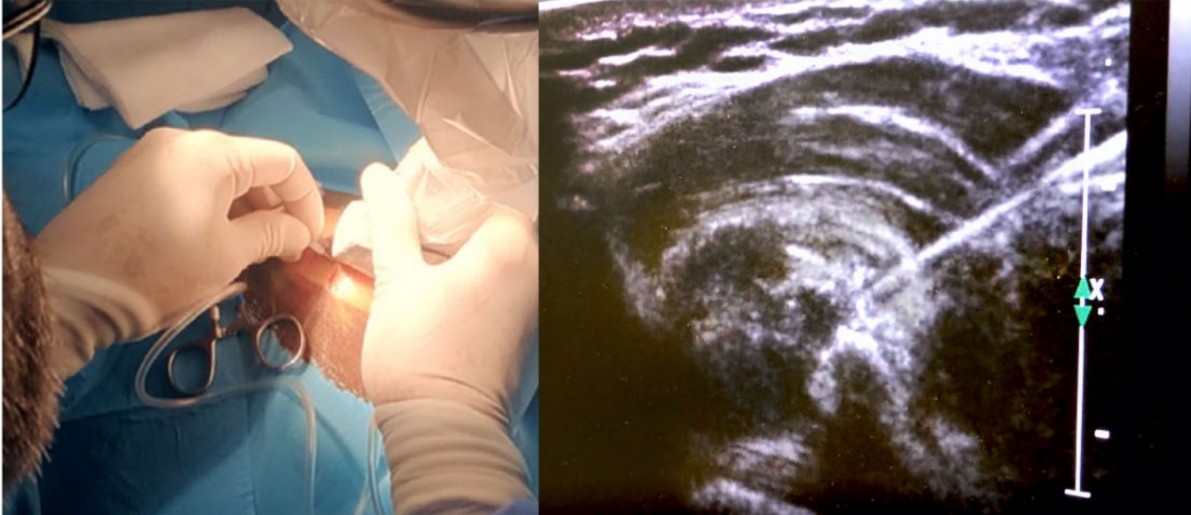

**Figure 5.** The third needle, usually of 22 gauge, is inserted slightly lower than the other two to break up the remainingcalcifications. At the end of the procedure, 40 mg of depo-medrol and 3 mL of 2% lidocaine are inserted into the subacromial-deltoid bursa. The process can take up to 45 min.

## 4. Discussion

The clinical history of CT of the shoulder is characterised by an asymptomatic period that can last for years and acute pain during the resorption phase.

Numerous studies confirm our initial hypothesis that the basis of the disease is functional overload, which leads to an inflammatory tendinopathy and a modification of gene expression with a subsequent metaplasia of tendon stem cells (TSC) into non-tenocytic cells such as chondrocytes, osteocytes or adipocytes [18,22–25] and a deposition of hydroxyapatite crystals.

Specifically, stretching greater than 8% [23] and high mechanical stresses result in a differentiation of TSCs into nontenocytic lineages (adipocytes, chondrocytes or osteocytes) and a deposition of HA crystals probably through a change in gene expression [17], release of epidermal growth factor (EGF) [21] and increased intracellular Ca2+ [31].

Thereafter, the local inflammatory state and subsequent reduction inthe extracellular pH (acidosis) results in the degradation of hydroxyapatite crystals by macrophages and multinucleated giant cells and a removal of calcium deposits [26,34], according to the following equation: Ca5 (PO4) 3 (OH) (s) $\rightleftarrows$ 5 Ca2+ + 3 PO43- + OH. This equilibrium is influenced by the pH value, as in the presence of acidosis, a subtraction of OH- ions from the dissociation equilibrium occurs, leading to the initiation of the hydrolytic process. Washing with a 0.9% sodium chloride solution allows the pH to return to a physiological value and for this reason, when performed at this stage, achieves maximum effectiveness as a pH buffer.

For this reason, it is essential to know the biochemical mechanisms that regulate the calcification cycle and calcification consistency, as they influence the choice of treatment, indications, timing and allow for the prediction of treatment efficacy.

In particular, based on our clinical experience, we consider the three-needle technique to be preferable as it allows the aspiration of more calcific content than the two-needle technique, which also facilitates the removal of more resistant calcific formations.

**Author Contributions:** Conceptualization, S.G.; methodology, S.M.S. and D.C.; formal analysis, S.M.S. and D.D.; data curation, F.V.; writing—Original draft preparation, M.M.; writing—Review and editing, S.G., D.B. and A.F.; supervision, A.F. All authors have read and agreed to the published version of the manuscript.

**Funding:** This research received no external funding.

**Institutional Review Board Statement:** Not applicable.

**Informed Consent Statement:** Not applicable.

**Data Availability Statement:** Not applicable.

**Conflicts of Interest:** The authors declare no conflict of interest.

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
