# Peer review of "Calcific Shoulder Tendinopathy (CT): Influence of the Biochemical Process of Hydrolysis of HA (Hydroxyapatite) on the Choice of Ultrasound-Guided Percutaneous Treatment (with the Three-Needle Technique)"

_2673-4036, doi:10.3390/osteology2030013_

Round 1

Reviewer 1 Report

Comments and Suggestions for Authors

An interesting paper in the Expert Opinion section. Calcific shoulder tendinopathy is very common in clinical practice, but till now seldom do I read papers discussing the influence of the biochem-2 ical process of hydrolysis of HA (hydroxyapatite) on the choice 3 of ultrasound-guided percutaneous treatment. This is a good reference for readers.

On the other hand, I think that the numbering of sections is not continuous. There is 1. Introduction, 2. Exposition of the theory and description of the personal infiltrative technique, 4. Discussion. Where is 3? Please add.

Author Response

Dear colleague,

We appreciate your helpful suggestions.

We confirm that we have made an error in paragraph numbering and have corrected it.

Reviewer 2 Report

Comments and Suggestions for Authors

Dear Authors,

I was very interested to read your manuscript as I do treat this condition in my practice (orthopaedics). The purpose of your paper is to report on a technique and rationale for this technique. This is an interesting manuscript, but somewhat difficult to read. There are too many 1-sentence paragraphs. Many sentences are strange. May I suggest that you find someone who can revise the sentences so they are more meaningful and impact? We lose the focus of the information you want to convey as some sentences are put there with no link to one another.

Your paper would also be much stronger and impactful if you could report some results of a proper sample of patients. Making it just as an "Expert Opinion" may be interesting to read, but not meaningful. For example, you say at line 106 that the three-needles technique is preferable, but you really did not study this with a proper sample of patients and a control group. So you cannot say that this method is preferable without the proper data to back this affirmation.

I also noticed that very many (if not most) of the articles in your reference list are quite old and not that recent. I would think that there must be more recent literature than early 2000 and before.

A few more things do not work: you say at line 95 that this technique is "alway indicated...", but in fact, there are contraindications. Line 240, you mention that "... the basis of the disease is functional overload, especially in athletes..." but in reality, athletes rarely have this condition and you said in lines 38-39 that this happens mostly in women in their 4th and 5th decades.

Ultimately, I believe you need to be careful with what you say and report and you would benefit from revising the entire manuscript to make it publishable.

Author Response

Dear colleague,
We appreciate your helpful suggestions to make our work more interesting and suitable for publication.
We have changed the text and content, improving the English and better explaining why, in our experience, the three-needle technique is preferable.
We have also cited more recent studies on calcific rotator cuff tendinopathy.

In the future, it will certainly be very interesting to conduct an RCT which compares the effectiveness of the two-needle technique with the three-needle technique.

Round 2

Reviewer 2 Report

Comments and Suggestions for Authors

Dear Authors,

I was happy to see a revised version of your manuscript. Unfortunately, most of my recommendations have not been followed. Therefore, in my opinion, your paper is still not ready for publication. Please see my comments and recommendations in the attached document. Most of those points are easy to edit and are minor and they will make your paper easier to read. But there are still a few important points. I was happy to see that you made efforts to update your list of references, but most of them are still quite old. Is there any way you can update this list further? Please see the attached document for all my comments for revisions.

Author Response

Dear colleague,
We appreciate your helpful suggestions.

We have consulted the attached document and made the required changes.  We have also further updated the list of references.
